# Cyproterone Acetate Mediates IRE1α Signaling Pathway to Alleviate Pyroptosis of Ovarian Granulosa Cells Induced by Hyperandrogen

**DOI:** 10.3390/biology11121761

**Published:** 2022-12-04

**Authors:** Yan Zhang, Xianguo Xie, Yabo Ma, Changzheng Du, Yuan Jiao, Guoliang Xia, Jinrui Xu, Yi Yang

**Affiliations:** 1Key Laboratory of Ministry of Education for Conservation and Utilization of Special Biological Resources in the Western, School of Life Sciences, Ningxia University, Yinchuan 750021, China; 2Reproductive Medicine Center, Yinchuan Maternal and Child Health Hospital, Yinchuan 750021, China; 3The Second Clinical College of China Medical University, 77 Puhe Road, Shenbei New District, Shenyang 110000, China

**Keywords:** hyperandrogenemia, pyroptosis, granulosa cells, IRE1α signaling pathway

## Abstract

**Simple Summary:**

Hyperandrogenemia (HA) is the main pathophysiological change that takes place in polycystic ovary syndrome (PCOS). Cyproterone acetate (CYA) is a drug commonly used to reduce androgen in patients with PCOS. Long-term and continuous exposure to hyperandrogen can cause ovarian granulosa cells (GCs), pyroptotic death, and follicular dysfunction in PCOS mice. The aim of this study was to investigate whether CYA could ameliorate HA-induced pyroptosis of PCOS ovarian GCs by alleviating the activation of the IRE1α signaling pathway. The results of this study showed that HA induced the pyroptosis of ovarian GCs by activating the IRE1α signaling pathway in mouse ovarian granulosa cells and KGN cells. CYA can alleviate the occurrence of ovarian GC pyroptosis by inhibiting the activation of the IRE1α signaling pathway. This study provides a new mechanism and evidential support for CYA in the treatment of PCOS patients.

**Abstract:**

Objective: Hyperandrogenemia (HA) is the main pathophysiological change that takes place in polycystic ovary syndrome (PCOS). Cyproterone acetate (CYA) is a drug commonly used to reduce androgen in patients with PCOS. Long-term and continuous exposure to HA can cause ovarian granulosa cells (GCs), pyroptotic death, and follicular dysfunction in PCOS mice. The aim of this study was to investigate whether CYA could ameliorate the hyperandrogenemia-induced pyroptosis of PCOS ovarian GCs by alleviating the activation of the IRE1α signaling pathway. Methods: Firstly, thirty PCOS patients with HA as their main clinical manifestation were selected as the study group, and thirty non-PCOS patients were selected as the control group. The GCs and follicular fluid of the patients were collected, and the expression of pyroptosis-related proteins was detected. Secondly, a PCOS mouse model induced by dehydroepiandrosterone (DHEA) was constructed, and the treatment group model was constructed with the subcutaneous injection of cyproterone acetate in PCOS mice. The expression of pyroptosis-related protein in ovarian GCs was detected to explore the alleviating effect of CYA on the pyroptosis of ovarian GCs in PCOS mice. Thirdly, KGN cells-i.e., from the human GC line-were cultured with dihydrotestosterone, CYA, and *ERN1* (IRE1α gene) small interfering RNA in vitro to explore whether CYA can alleviate the activation of the IRE1α signaling pathway and ameliorate the hyperandrogenemia-induced pyroptosis of PCOS ovarian GCs. Results: The expression of pyroptosis-related proteins was significantly increased in ovarian GCs of PCOS patients with HA as the main clinical manifestation, and in the PCOS mouse model induced by DHEA. After treatment with CYA, the expression of pyroptosis-related proteins in the ovarian GCs of mice was significantly lower than that in PCOS mice. In vitro experiments showed that CYA could ameliorate KGN cells’ pyroptosis by alleviating the activation of the IRE1α signaling pathway. Conclusion: This study showed that CYA could ameliorate the activation of the IRE1α signaling pathway in mouse GCs and KGN cells, and also alleviate pyroptosis in ovarian GCs. This study provides a new mechanism and evidential support for CYA in the treatment of PCOS patients.

## 1. Introduction

Polycystic ovary syndrome (PCOS) is a common reproductive endocrine disease in women of childbearing age. The incidence rate is about 9~18% and, significantly, as many as 80% of women with ovulation disorder infertility suffer from PCOS [1,2]. The main clinical manifestations of PCOS include infertility, menstrual disorder, hirsutism, obesity, acne, and so on. Hyperandrogenemia (HA) is the main pathophysiological basis of PCOS. Some studies have shown that excessive androgen in PCOS patients can promote the recruitment and growth of follicles, producing a large number of small and immature follicles, eventually leading to polycystic changes in the ovary. Conversely, when the follicle diameter reaches 2~8 mm, the increased androgen will promote atresia [3,4]. At the same time, these small follicles also secrete excessive levels of anti-Mullerian hormone to inhibit the aromatase activity induced by follicle-stimulating hormone (FSH) and hinder follicular development. This leads to ovarian dysfunction [5]. It has also been reported that excessive androgen can induce low level inflammatory reaction in PCOS patients, forming a vicious cycle that ultimately leads to follicle development disorder [6,7].

The developmental ability of oocytes to undergo meiosis, as well as fertilize and form healthy embryos, is one of the key factors determining female fertility. Ovarian granulosa cells (GCs) are the somatic cells around oocytes, and are the transport channels for nutrients, regulatory molecules, and paracrine factors during oocyte development and maturation. Therefore, ovarian GCs play a crucial role in oocyte development. Studies have shown that abnormal ovarian GC function plays a key role in follicle development and chronic anovulation [8]. Professor Long Shuanglian’s team found that androgen upregulated the expression of ovarian klotho, induced the apoptosis of GCs, and promoted the occurrence and development of PCOS. In a study of ovarian GCs in PCOS patients, Professor Songying Zhang’s team also found that androgen activated endoplasmic reticulum stress (ERS) in mouse ovarian GCs through the activation of the p38 MAPK signaling pathway, and then caused the excessive response of GCs to LH, leading to the abnormal function of ovarian GCs [9]. Recent studies have also pointed out that HA in PCOS mice leads to ovarian GC pyroptosis through the activation of the NLRP3 inflammasome, and ultimately leads to follicle dysfunction [10].

In clinical treatment, drugs should be used to reduce androgen levels in PCOS patients with HA as the main clinical manifestation. Cyproterone acetate (CYA) is a progestin-based drug that can be combined with estradiol to form a compound oral preparation used for contraception, menstrual disorders, hormone replacement therapy for menopausal syndrome, and female androgen-dependent diseases, such as seborrheic dermatitis, acne, PCOS, etc. If cyproterone acetate is used alone, it is generally used to reduce male sexual desire and severe masculinity in women. It can also be used as a conservative treatment for prostate cancer [11]. At present, there is no relevant study on whether ovarian GC pyroptosis exists in PCOS patients with HA as the main clinical manifestation, or whether CYA treatment can alleviate ovarian GC pyroptosis and its mechanism. This study will offer new insights into the mechanism by which CYA regulates the hyperandrogen-induced pyroptosis of ovarian GCs in PCOS mice, and provide new ideas and theoretical bases for alleviating the status of ovarian GCs to ameliorate oocyte quality in clinical practice.

## 2. Materials and Methods

### 2.1. Materials

#### 2.1.1. Source of Human Specimens

Inclusion criteria: A total of 60 patients who had received assisted reproductive treatment in the Reproductive Center of Yinchuan Maternal and Child Health Hospital were selected. Thirty patients with PCOS were chosen as the experimental group, and thirty healthy women who had received assisted reproductive treatment due to male infertility were included as the control group. All patients signed informed consent and were approved by the Ethics Committee of Yinchuan Maternal and Child Health Hospital. PCOS patients were included according to the Rotterdam diagnostic criteria published in 2003 [12]. They had to meet the requirements of hyperandrogenism or clinical manifestations of hyperandrogen in this study, as well as one of the following: (a) sparse ovulation or anovulation; (b) gynecological ultrasound showing ovarian polycystic changes (≥12 follicles with a diameter of 2–9 mm and/or an ovarian volume of ≥10 mL in one or both ovaries). The inclusion criteria of the control group were as follows: (a) ovarian morphology confirmed to be normal by ultrasound examination; (b) menstrual cycles of 28–32 days; (c) a normal endocrine hormone level during the 3–5 days of the menstrual cycle.

The exclusion criteria were as follows: (a) age > 39 or <21 years; (b) a history of previous ovarian surgery or disease; (c) other factors of hyperandrogenism, such as congenital adrenal hyperplasia, hyperprolactinemia, thyroid diseases, Cushing’s syndrome, etc.

Serum was collected on the second day of menstruation. On the day of oocyte retrieval, follicular fluid and granulosa cells were collected and frozen at −80 °C. In addition, the ovarian GCs of the two groups were collected and cultured for 24 h, and then fixed with 2.5% glutaraldehyde fixation solution and 4% paraformaldehyde for electron microscopy and cell immunofluorescence detection, respectively.

#### 2.1.2. Source of Animal Specimens 

The C57BL/6 strain mice used in the experiment were purchased from Ningxia Medical University, with an age of about 4 weeks and a weight of about 15 g. All animals used in the laboratory passed the resolution of the Animal Ethics Committee of Ningxia University (ethics approval no. NXU-2020-012).

The mouse models of the PCOS group, the control group, and the treatment group were constructed: five mice in the PCOS group were injected subcutaneously with dehydroepiandrosterone (DHEA) for 21 days. The required injection concentration of DHEA was 6 mg/100 g, the concentration of DHEA was set to 12 mg/mL, and DHEA was dissolved in Cremophor EL. Five mice in the control group were injected subcutaneously with the same dose of Cremophor EL as the PCOS group for 21 days. Five mice in the treatment group were injected subcutaneously with DHEA and cyproterone acetate (CYA) for 21 days. The required concentration of CYA was 2 mg/100 g, and the concentration of CYA was set at 4 mg/mL, which was also dissolved in Cremophor EL. The feeding conditions of all mice were: a constant temperature of 22 °C; humidity of 50~60%; light/dark cycle of 12/12 h; and sufficient food and drinking water.

After successful modeling, blood samples were collected from the mice, and the serum was separated and stored at −80 °C to measure the total testosterone, IL-18, and IL-1β. Bilateral ovaries were obtained, and one side was frozen at −80 °C for protein extraction in subsequent tests. The other side was fixed in 4% paraformaldehyde, and tissue embedding and sectioning were performed for the H&E and immunohistochemical tests.

#### 2.1.3. Source of Vitro Specimens

KGN cells, i.e., the human granulosa cell line, were used as the experimental material. After culture and adherence, the drug addition experiment could be started when the culture medium was changed for 1–2 days to about 70% cell density.

The cell models of the DHT group, control group, and treatment group were established, and DHT solution (DHT dissolved in DMSO at a concentration of 10 μM) was added to the DHT group for 24 h. DMSO solution with the same volume as the DHT group was added to the control group for 24 h. Then, 10 μM DHT solution was added to the treatment group, and after 2 h of action, CYA solution was added (CYA dissolved in DMSO in concentration of 10 μM) for 22 h.

The cell models of the DHT group, the control group, the si-*ERN1* group, and the si-*ERN1*+DHT group were established. The si-*ERN1* was transfected into KGN cells to knockdown *ERN1* expression in the si-*ERN1* group and the si-*ERN1*+DHT group. DHT solution was added to the DHT group and the si-*ERN1*+DHT group for 24 h, and DMSO solution with the same volume as the DHT group was added to the control group and the si-*ERN1* group for 24 h. The cells of each group were collected, parts of the cells were fixed with 4% paraformaldehyde for immunofluorescence detection, and the other parts of the cells and cell supernatant were stored at −80 °C for the subsequent detection of various indicators.

### 2.2. Methods 

#### 2.2.1. Western Blotting

SDS-page (Yamei, PG113) glue was prepared with an appropriate concentration, and sample loading, electrophoresis, and membrane transformation were performed. After membrane transformation, PVDF (Millipore, R1JB38278) membrane containing the target protein was placed in blocking solution containing 5% skim milk (BD, 1053907) for 1 h at room temperature. The primary antibodies (Grp78, CST, #3177; TXNIP, abcam, ab188865; NlRP3, CST, #15101; cleaved GasderminD, CST, #36425; p-IRE1α, abcam, ab124945; cleaved caspase-1, Affinity, AF4005; GasderminD, CST, #46451; caspase-1, CST, #24232; IRE1α, CST, #3294) were incubated at 4 °C overnight, and PBST (Tween 20: Solarbio, T8220, PBS: Servicebio, G0002) was washed 3 times, for 5 min each time. The membranes were incubated with horseradish peroxidase-conjugated secondary antibodies (Proteintech, SA00001-1/SA00001-2). Then, PBST was washed 3 times, for 5 min each time, and ECL (ThermoFisher, XA338899) chemiluminescence reagent was prepared.

#### 2.2.2. Immunohistochemistry

Ovaries were fixed in 4% paraformaldehyde (Solarbio, P1110) overnight at 4 °C. Samples were cut into 5 μm thick sections, transferred to 3-aminopropyl triethoxysilane-treated slides (ZSBB-Bio, ZLI-9001), dehydrated in ethanol and toluene, embedded in paraffin, and stained for IHC according to the instructions of the IHC Kit (ZSBB-Bio, PV-9001). DAB reagent was used to detect antibody complexes according to the manufacturer’s instructions.

#### 2.2.3. Enzyme-Linked Immunosorbent Assay

The levels of IL-18, IL-1β, and total testosterone in serum, follicular fluid, and cell supernatant were detected by ELISA kit (Shanghai Enzyme-Linked Biotechnology Co., LTD., Shanghai, China, YJ112581). The detection method was carried out according to the kit instructions. In brief, the samples were added to the bottom of the plate well and incubated at 37 °C for 30 min. Each well was filled with washing solution, left to stand for 30 s, and then discarded. This was repeated 5 times. Then, 50 μL HRP-conjugated reagent was added to each well, except for the blank well, and incubated at 37 °C for 30 min. Each well was filled with washing solution, left to stand for 30 s, and then discarded. This was repeated 5 times. Then, 50 μL chromogenic agent A was added to each well, followed by 50 μL chromogenic agent B. Gentle shaking and mixing took place at 37 °C, avoiding light for 10 min. The reaction was terminated by adding 50 μL stop solution. The absorbance of each well was measured sequentially at a wavelength of 450 nm.

#### 2.2.4. Hematoxylin–Eosin (H&E) Staining

Ovarian tissues were embedded in paraffin and processed on slides for H&E staining in order to examine the pathological structural alterations of the ovary.

#### 2.2.5. Cellular Immunofluorescence

For GC staining, sections were left in 4% paraformaldehyde for 30 min at 25 °C, and then permeabilized with 0.3% Triton X-100. After washing with PBS three times, cells were blocked with 3% BSA for 30 min at 25 °C; then, cells were incubated with antibodies against NLRP3 (1:100, Affinity, DF7438), p-IRE1α (1:100, Affinity, AF7150), GasderminD (1:100, Affinity, AF4012), and caspase-1 (1:100, αAffinity, AF5418) overnight at 4 °C. The next day, after washing with PBS three times, cells were incubated with Alexa Fluor 488-conjugated secondary antibodies (1:100, Yeasen, 34206ES60) for 1 h at 37 °C and Hoechst 33,342 (ZSGB-BIO, C1028) as a nuclear counterstain. Then, samples were observed under a microscope.

#### 2.2.6. Transmission Electron Microscopy

Samples were prefixed using 2.5% glutaraldehyde and postfixed in 1% osmium tetroxide, followed by dehydration in an acetone series, Epox 812 infiltration, and embedding. Semi-thin sections were then stained using methylene blue, and ultra-thin sections were prepared using a diamond knife, followed by uranyl acetate and lead citrate staining. A JEM-1400-FLASH transmission electron microscope (JEOL, Tokyo, Japan) was then used to image sections.

#### 2.2.7. Scanning Electron Microscopy

After sampling, samples were prefixed using 2.5% glutaraldehyde, and then washed twice with water for 5 min each time and dehydrated with a series of gradients of alcohol at 30%, 50%, 70%, 80%, 90%, 95%, and 100% (10 min for each gradient). The samples were gently glued with conductive glue, and then sprayed via ion sputtering. Finally, the samples were observed at the appropriate position with the appropriate multiple under the microscope.

#### 2.2.8. Small Interfering RNA Interference Assay in KGN Cells

KGN cells were first starved with DMEM-F12 medium without FBS for 12 h after culture adherence. Then, according to the instructions of the Lipofectamine TM3000 transfection reagent, si-*ERN1* was mixed with Lipofectamine TM3000 at 10 pmol. The proportion of 1 μL was added to serum-free OPTI-MEM, was mixed with full shaking, and left at room temperature for 15 min. Finally, the above mixture was added to FBS-free DMEM-F12 medium, and then added to the cell culture well plate. After transfection for 6 h, the medium containing FBS was changed and continued to be cultured for 24 h for the subsequent experiments.

#### 2.2.9. Statistical Analysis

The experimental results were obtained from at least three or more repeated experiments. The Image J 1.8.0 software was used to analyze the gray values of the Western blot results. GraphPad Prism 8.0.2 analysis software was used for comparison, and SPSS22.0 software was used for the statistical analysis of clinical data. Measurement data with normal distribution were expressed as mean ± standard deviation (mean ± SD), and comparison between groups was performed via independent sample t test. The adoption rate of count data (%) was expressed, and comparison between groups was performed with the χ^2^ test; *p* < 0.05 indicates statistical significance. (Additional data used in the analysis please see Appendix A.)

## 3. Results

### 3.1. HA Leads to Ovarian GC Pyroptosis in PCOS Patients through the Activation of the NLRP3 Inflammasome

Recent studies have also pointed out that HA in PCOS mice leads to ovarian GC pyroptosis through the activation of the NLRP3 inflammasome, and ultimately leads to follicle dysfunction. Thirty PCOS patients with HA as the main feature and thirty control patients were included in this study. The basic information and pregnancy outcomes of the two groups were compared. At the same time, the follicular fluid and GCs of the two groups were collected for the detection of total testosterone (TT), inflammatory factors, and pyroptosis-related proteins. The level of TT in serum, the number of antral follicles, and the number of oocytes captured were higher in the PCOS group, but the number of mature oocytes was similar between the two groups. The number of excellent embryos and clinical pregnancy rate were significantly lower, but the early abortion rate was higher in the PCOS group (Table 1). This suggests that the quality of oocytes and embryos decreased in the PCOS group. The levels of TT, IL-18, and IL-1β were significantly higher in the follicular fluid of the PCOS group (Figure 1A). The expression of the NLRP3 inflammasome in the ovarian GCs of PCOS patients was significantly higher (Figure 1B,C). This indicates that the ovary was in a state of HA and chronic inflammation. The GCs of the two groups were observed via transmission electron microscopy and scanning electron microscopy. Scanning electron microscopy images showed that, compared with the control group, the GCs of the PCOS patients were significantly swollen, the granular uplift on the cell surface disappeared, and there were many tiny pores with a dense distribution. As indicated by the red arrow (Figure 1D), transmission electron microscopy showed that the membranes of the GCs of the PCOS patients expanded outward and ruptured, with nuclear pyknosis, nucleolus disappearance, heterogeneous chromatin fragmentation, lysosome swelling, mitochondrial number reduction, and a loss of normal structural appearance. As indicated by the red arrow (Figure 1E), the morphological changes in the GCs of the PCOS patients were consistent with the pyroptosis characteristics reported in the literature by electron microscopy [13]. Pyroptosis-related proteins were detected in the GCs of the two groups. The expressions of GSDMD and caspase-1 and their cleaved fragments were also significantly increased in the PCOS group (Figure 1F,G). These results suggest that HA can lead to the pyroptosis of ovarian GCs by activating the NLRP3 inflammasome in PCOS patients. Ovarian GC pyroptosis can promote the production and secretion of inflammatory factors such as IL-18 and IL-1β, aggravate a local chronic inflammatory response in the ovary, and ultimately affect oocyte quality and pregnancy outcome.

### 3.2. CYA Can Effectively Alleviate Ovarian GC Pyroptosis by Reducing HA

In the experiment described above, it was found that the hyperandrogen activation of NLRP3 caused the pyroptosis of ovarian GCs in patients with PCOS. CYA is an androgen-lowering drug commonly used in patients with PCOS. In order to explore whether CYA can effectively reduce androgen levels and alleviate the occurrence of ovarian GC pyroptosis in PCOS patients, three animal models were constructed in this study—the PCOS group, the treatment group and the control group—to detect the expression of pyroptosis-related proteins, TT, IL-18, IL-1β, and the NLRP3 inflammasome. The study found that the estrous cycle of mice was regular in the control group: there were two complete estrous cycles on average within two weeks (Figure 2A,B). In contrast, mice in the PCOS group remained in estrus, indicating estrous cycle disorder (Figure 2A,B). HE counts in ovarian slices showed that the number of cystic follicles in PCOS mice was significantly higher than that in the control group, while luteal cysts and preovulatory follicles were significantly lower than those in the control group. This indicated that there were polycystic changes and sparse ovulation in the ovaries of the PCOS mice (Figure 2D). The TT in serum was significantly higher than that of the control group (Figure 2C), which indicated that the PCOS mouse model was successfully constructed. CYA belongs to the progesterone drug, so the estrous cycle of the mice was always in the diestrus phase of the treatment group (Figure 2A,B), the cystic follicles of the mice of the treatment group were significantly less than those of the PCOS group (Figure 2D), and the serum TT of the treatment group was also significantly lower than that of the PCOS group (Figure 2C). This indicated that the mouse model of the treatment group was successfully constructed. 

By detecting the expression levels of pyroptosis-related proteins such as GSDMD, caspase-1, and their cleaved fragments in the ovaries of the three groups of mice, it was found that the expression levels of pyroptosis-related proteins in the ovaries of PCOS mice were significantly higher than those in the control group, and the expression level of pyroptosis proteins was significantly lower in CYA-treated mice than in the PCOS group (Figure 3B,C). The expression levels of IL-18, IL-1β, and the NLRP3 inflammasome in the serum of the three groups of mice were detected at the same time, and the expression levels of IL-18, IL-1β, and the NLRP3 inflammasome in PCOS mice were significantly higher than those in the control group. The expression levels of IL-18, IL-1β, and the NLRP3 inflammasome were significantly lower in CYA-treated mice than in the PCOS group (Figure 3A,C). These results suggest that CYA alleviates ovarian GC pyroptosis by reducing the androgen level in PCOS mice, and then alleviates the release of inflammatory factors IL-18 and IL-1β caused by pyroptosis and inhibits the local chronic inflammatory response in the ovaries.

### 3.3. CYA Can Effectively Alleviate the Activation of the IRE1α Signaling Pathway in Ovarian GCs by Reducing Androgen Levels

The IRE1α pathway is the most evolutionarily conserved branch of the unfolded protein response (UPR) signaling network. Thioredoxin-interacting protein (TXNIP) downstream of the IRE1α pathway has been proved to regulate pyroptosis through the TXNIP/NLRP3 pathway in acute kidney injury, hepatocellular injury, preeclampsia, and other diseases. Therefore, the IRE1 pathway was first selected for investigation in this study. IRE1α signaling pathway proteins were detected in the ovaries of the three groups of mice, and it was found that the expressions of GRP78, IRE1α, p-IRE1α, and TXNIP were significantly higher in the PCOS mice than those in the control group (Figure 4A,B). The expressions of IRE1α signaling pathway proteins were significantly lower in CYA-treated mice than those in the PCOS group (Figure 4A,B). These results suggest that CYA can effectively alleviate the activation of the IRE1α signaling pathway in ovarian GCs by reducing androgen levels in PCOS mice. 

### 3.4. CYA Could Effectively Alleviate the Activation of the IRE1α Signaling Pathway and Pyroptosis in Ovarian GCs Induced by Hyperandrogen in In Vitro Experiments

To further investigate the effect of CYA on IRE1α signaling pathway activation and pyroptosis in ovarian GCs induced by hyperandrogen, this study conducted in vitro experiments. The KGN cells from the ovarian granulosa cell line were used as the research object, and control, DHT, and DHT + CYA groups were set up. The expression of IRE1α signaling pathway proteins was detected in the three groups, and it was found that the expressions of GRP78, IRE1α, p-IRE1α, and TXNIP were significantly higher in the DHT group than in the control group. After treatment with CYA, the expression levels of the IRE1α signaling pathway proteins were significantly lower than those in the PCOS group (Figure 5A,B). The expression levels of pyroptosis-related proteins such as GSDMD, caspase-1, and their cleaved fragments were detected in the three groups. It was found that the expression levels of pyroptosis-related proteins in the DHT group were significantly higher than in the control group, and after treatment with CYA, the expression levels of pyroptosis-related proteins were significantly lower than those in the DHT group (Figure 5D,E). The expression levels of IL-18, IL-1β, and the NLRP3 inflammasome in the cell supernatants and GCs of the three groups were detected. The results showed that the expression levels of IL-18, IL-1β, and the NLRP3 inflammasome in the DHT group were significantly higher than those in the control group, and after treatment with CYA, the expression levels of IL-18, IL-1β, and the NLRP3 inflammasome were significantly lower than those in the DHT group (Figure 5C,E). These results suggested that CYA could effectively alleviate the activation of the IRE1α signaling pathway and pyroptosis in ovarian GCs in the DHT group.

### 3.5. CYA Can Alleviate Ovarian GC Pyroptosis and Local Ovarian Inflammatory Response by Inhibiting the Activation of the IRE1α Signaling Pathway in PCOS

The above experiments show that CYA can effectively alleviate ovarian GC pyroptosis caused by HA. The expression of IRE1α signaling pathway proteins declined in the in vitro experiment, and thus it is speculated that CYA could ameliorate the activation of the IRE1α signaling pathway and alleviate ovarian GC pyroptosis. In order to prove this, *ERN1*-which is a IRE1α gene-was knocked down in KGN cells by siRNA, and the control group, the DHT group, the si-*ERN1* group, and the si-*ERN1*+DHT group were set up. The expressions of IRE1α signaling pathway proteins and pyroptosis-related proteins were detected in each group. GRP78-which is an upstream protein of IRE1α-was not affected by si-ERN1. The expression of GRP78 was significantly higher in the si-*ERN1*+DHT group and the DHT group, but there was no significant difference between the si-*ERN1*+DHT group and the DHT group (Figure 6A). The expression levels of IRE1α, p-IRE1α, and TXNIP in the si-*ERN1*+DHT group were significantly lower in the si-*ERN1*+DHT group than in the DHT group (Figure 6A,B), and the expression levels of the NLRP3 inflammasome and pyroptosis-related proteins such as GSDMD, caspase-1, and their cleaved fragments were also decreased in the si-*ERN1*+DHT group, compared to those in the DHT group (Figure 6D,E). The expressions of IL-18 and IL-1β were detected in the cell supernatants of the four groups. The expression levels of IL-18 and IL-1β in the si-ERN1+DHT group were significantly lower than those in the DHT group (Figure 6C). It is confirmed that CYA can alleviate ovarian GC pyroptosis and local inflammatory response by inhibiting the activation of the IRE1α signaling pathway in PCOS (Figure 7).

## 4. Discussion

### 4.1. HA Induces Ovarian GC Pyroptosis by Activating the NLRP3 Inflammasome in PCOS Patients

Pyroptosis is a type of programmed cell necrosis mediated by gasdermin, which is characterized by cell swelling and rupture, the release of contents, and a strong inflammatory reaction. It plays an irreplaceable role in resisting the invasion of external pathogens and sensing endogenous danger signals [13,14,15]. At present, many diseases have been found to be accompanied by pyroptosis, such as infectious diseases, gout, immune deficiency diseases, autoimmune diseases, and so on [16,17,18]. Studies have shown that in patients with diabetic nephropathy, hyperglycemia can activate NLRP3, which coexists with ASC and pro-caspase-1 to form the NLRP3 inflammasome, and then activates caspase-1. Active caspase-1 promotes the maturation and secretion of pro-IL-1β, and cleaves GSDMD to induce the occurrence of pyroptosis [19,20,21]. Recent studies have also shown that HA induces ovarian GC pyroptosis through the activation of the NLRP3 inflammasome in PCOS mice, and then promotes the secretion of inflammatory factors, causing follicle development dysfunction [11]. In this study, we collected the serum, follicular fluid and ovarian GCs of PCOS patients to detect relevant indicators. It was found that the expression level of the NLRP3 inflammasome in the ovarian GCs of PCOS patients was significantly higher than in the control group. At the same time, the detection of caspase-1 and GSDMD and their shear fragments of ovarian GC pyroptosis also indicated that the related proteins of ovarian GC pyroptosis in PCOS patients were significantly higher than those in the control group, and the expression levels of IL-18 and IL-1β in follicular fluid were also significantly higher than those in the control group. The comparison and analysis of the basic data and pregnancy outcome of the two groups showed that although the number of oocytes obtained was more than that of the control group, the number of mature oocytes of PCOS patients was similar to that of the control group. Thus, it can be seen that the rate of oocyte maturity of PCOS patients was lower. At the same time, the rate of excellent embryos and clinical pregnancy were lower in the PCOS group than in the control group, and the abortion rate was higher than that in the control group. It is suggested that the outcome of pregnancy aid in PCOS patients is lower than the control group. Therefore, it is speculated that HA in PCOS patients can lead to the occurrence of pyroptosis of ovarian GCs by activating the NLRP3 inflammasome. Thus, pyroptosis could encourage ovarian GCs to secrete inflammatory factors such as IL-18 and IL-1β, leading to the occurrence and development of chronic inflammation in the ovaries of PCOS patients. The quality of oocytes and pregnancy outcome were negatively affected by ovarian GC pyroptosis and chronic inflammation.

### 4.2. CYA Can Effectively Alleviate Ovarian GC Pyroptosis and the Activation of the IRE1α Signaling Pathway of PCOS Induced by HA

Some factors, including hypoxia, oxidative stress, Ca2+ homeostasis, and others, in vivo and in vitro can lead to the accumulation of misfolded or unfolded proteins and induce endoplasmic reticulum stress (ERS). ERS will cause the separation and activation of three key proteins in the signal pathway—namely, inositol essential enzyme l (IREl), protein kinase R-like endoplasmic reticulum kinase (PERK), and active transcription factor 6 (ATF6)-from glucose-regulated protein 78 (GRP78), triggering a series of complex signal transmissions. This is called UPR [22,23,24]. UPR can reduce protein synthesis; however, it can also enhance protein folding ability, alleviate ERS, and maintain cell homeostasis. However, persistent or severe ERS can induce cell death [25,26]. Current studies have shown that ERS is widely present in ovarian injury and ovarian-related diseases. For example, it can cause a decrease in the progesterone secretion of GCs induced by inhibiting the steroidogenic enzyme of GCs in obese women, thus leading to luteal insufficiency [27,28]. Some studies have also shown that when embryos are cultured in vitro, oxides can negatively affect the development of blastocysts by mediating the expression of ERS-related molecules. ERS can also cause the apoptosis of GCs, decrease the mitochondrial function of oocytes, promote the apoptosis of oocytes, reduce the quality of oocytes, cause embryonic development to stop, and promote apoptosis [29,30]. IRE1α is a type I transmembrane protein, and its C-terminus is located in the cytosol. IRE1α has the dual activities of protein kinase and endonuclease [31,32]. When ER stress occurs, IRE1α dissociates from GRP78, dimerizes, and autophosphorylates. Its RNase activity is activated, which mediates the cleavage of XBP1 mRNA, and then translates into the transcription factor XBP1s. At the same time, p-IRE1α can regulate IRE1α-dependent decay, which degrades specific ER bound mRNA, such as the degradation of miR17 by p-IRE1α, and can promote the transcription and expression of TXNIP and alleviate the occurrence of ERS [33,34].

CYA is one of the main components of ethinylestradiol and cyproterone acetate tablets, which are commonly used in clinical contraceptives. It can effectively reduce androgen levels, so it is often used in PCOS patients with high androgen as the main clinical manifestation. It can effectively improve endocrine disorders and improve ovulation induction pregnancy rate in PCOS patients. However, whether the function of ovarian granulosa cells in PCOS patients can be improved after androgen reduction, or whether the quality of oocytes can be improved, has not been studied [35,36]. To investigate whether CYA can effectively alleviate ovarian GC pyroptosis and the activation of the IRE1α signaling pathway in PCOS induced by HA, we successfully established mice and cell models of PCOS and CYA treatment groups. The expression levels of IL-18 and IL-1β in the serum or cell supernatant of mice in each group were detected. The expression levels of the IRE1α pathway protein, the NLRP3 inflammasome, and pyroptosis-related proteins were detected in mice ovaries and KGN cells treated with drugs in each group. The study found that CYA can effectively reduce the expression of pyroptosis-related proteins in ovarian GCs, and the expression levels of IL-18, IL-1β, and the NLRP3 inflammasome also decreased significantly. It is suggested that CYA can also alleviate the chronic inflammatory reaction caused by HA, and it was found that the expression level of IRE1α pathway proteins in the treatment group was significantly lower than that in the PCOS group. In conclusion, CYA can effectively alleviate hyperandrogen-induced pyroptosis and chronic inflammatory response in ovarian GCs at both the animal and cellular levels, and it may be related to the inhibition of IRE1α pathway activation.

### 4.3. The Mechanism by Which CYA Effectively Alleviates the Hyperandrogen-Induced Pyroptosis of GCs Was Investigated

In order to explore the mechanism of CYA in relation to GC pyroptosis caused by HA, this experiment used siRNA to knockdown *ERN1* (IRE1α gene), and used a high-concentration androgen treatment. It was found that with the decrease in IRE1α pathway protein expression level, pyroptosis-related protein expression levels also decreased, and the expression levels of IL-18, IL-1β, and the NLRP3 inflammasome decreased significantly. In conclusion, CYA alleviates the occurrence of ovarian GC pyroptosis and chronic ovarian inflammatory response caused by hyperandrogen by alleviating the activation of the IRE1α pathway. It has been reported that thioredoxin-interacting protein (TXNIP) is a key node in the disruption chain from ERS to programmed cell death [37]. Studies on type II diabetes mellitus, acute kidney injury, hepatocyte injury, preeclampsia, and other diseases have shown that ERS can regulate pyroptosis through the TXNIP/NLRP3 pathway [38]. In addition, studies in PCOS rats have shown that the TXNIP/NLRP3 pathway can promote the occurrence of chronic ovarian inflammation [39]. Therefore, considering the above experimental results combined with the literature, it is speculated that IRE1α pathway activation, possibly through the TXNIP/NLRP3 pathway, activates Capase-1 and causes ovarian GC pyroptosis and chronic inflammation.

This study was limited to the study of ovarian granulosa cells at animal and cellular levels. In a later study, we will study oocytes to explore whether CYA can effectively improve the treatment of PCOS oocytes by improving ovarian granulosa cells, in turn improving pregnancy outcomes.

## 5. Conclusions

In conclusion, the present study provides evidence that CYA can alleviate hyperandrogen-induced ovarian GC pyroptosis and chronic ovarian inflammatory response by alleviating IRE1α pathway activation in vivo and in vitro.

## Figures and Tables

**Figure 1 biology-11-01761-f001:**
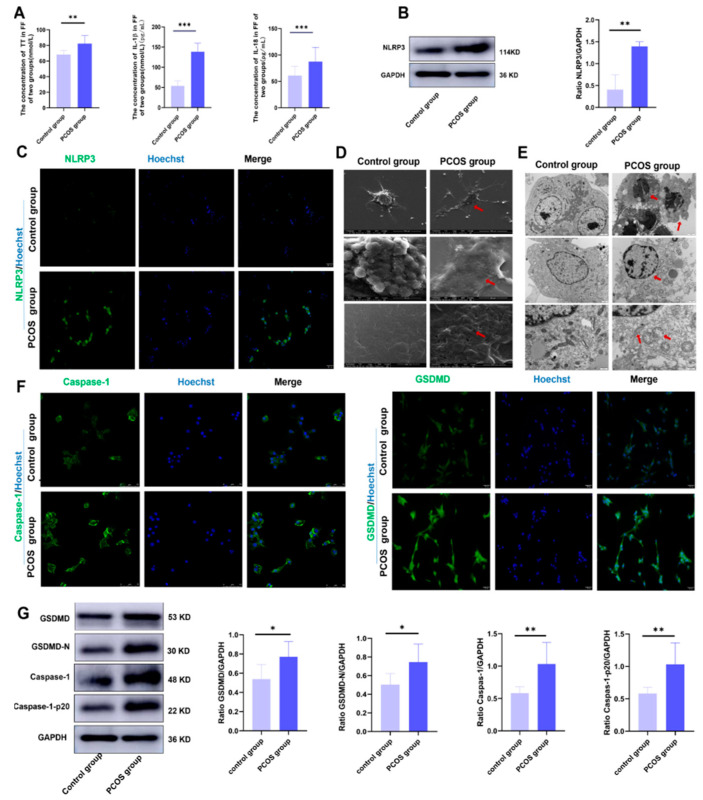
Hyperandrogenism causes pyroptosis of ovarian GCs by activating NLRP3 inflammasome in PCOS patients. (**A**) The expression of TT, IL-18, and IL-1β in follicular fluid were analyzed using enzyme-linked immunosorbent assay kits in the PCOS group and the control group; (**B**) the expression of NLRP3 inflammasome of ovarian GCs was analyzed with Western blot in the two groups; (**C**) the expression of NLRP3 inflammasome of ovarian GCs was analyzed with immunofluorescence staining of the two groups. The scale is 50 um. (**D**) The pyroptosis characteristics of ovarian GCs were observed with scanning electron microscopy; (**E**) the pyroptosis characteristics of ovarian GCs in the two groups were observed with transmission electron microscopy; (**F**) the expression levels of caspase-1 of ovarian GCs were analyzed with immunofluorescence staining. The scale is 75 um. The expression levels of GSDMD of ovarian GCs were analyzed with immunofluorescence staining. The scale is 50 um. (**G**) The expression levels of pyroptosis-related proteins of ovarian GCs were analyzed with Western blot. Data are shown as mean ± SEM. * *p* < 0.05, ** *p* < 0.01, *** *p* < 0.001.

**Figure 2 biology-11-01761-f002:**
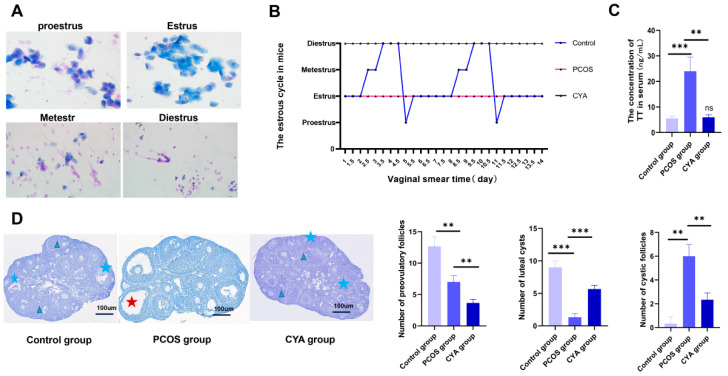
Three groups of mice were successfully constructed. (**A**) Schematic diagrams of vaginal smear in the three groups of mice (n = 5); (**B**) schematic diagrams of the estrus cycle in the three groups of mice (n = 5); (**C**) TT in serum was analyzed using enzyme-linked immunosorbent assay kits in the three groups of mice; (**D**) ovarian morphology in the three groups of mice was assessed with H&E staining, and ovulatory follicles, cystic follicles, and corpus luteum were counted; data are shown as mean ± SEM. “*” indicates represents comparison between groups, “ns” indicated that there was no statistical difference between the treatment group and the control group. ** *p* < 0.01, *** *p* < 0.001. 
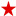
: the cystic follicle; 
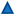
: the corpus luteum cyst; 
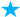
: the preovulatory follicle.

**Figure 3 biology-11-01761-f003:**
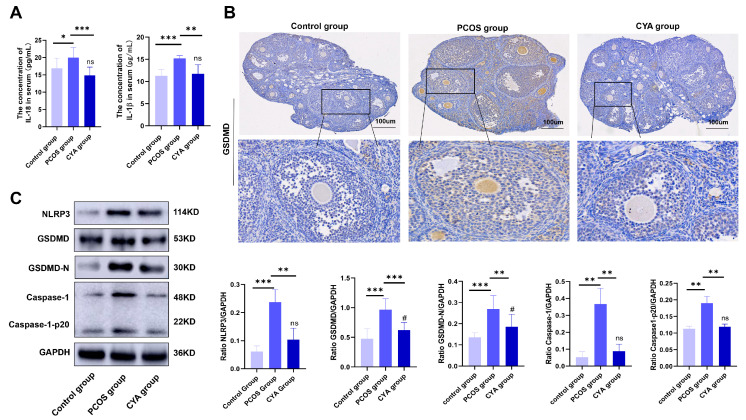
CYA can effectively alleviate the expression of inflammatory factors and ovarian GC pyroptosis-related proteins in PCOS mice induced by hyperandrogen. (**A**) The expression of IL-18 and IL-1β in serum was analyzed with enzyme-linked immunosorbent assay kits in the three groups of mice; (**B**) the expression of GSDMD in ovaries was assessed with immunohistochemical tests; (**C**) the expression of NLRP3 and pyroptosis-related protein in ovaries was assessed with Western blot. Two independent experiments were performed with similar results. Data are shown as mean ± SEM. “*” represents comparison between groups, “#” indicates the treatment group compared to the control group, “ns” indicated that there was no statistical difference between the treatment group and the control group. *^/#^
*p* < 0.05, ** *p* < 0.01, *** *p* < 0.001.

**Figure 4 biology-11-01761-f004:**
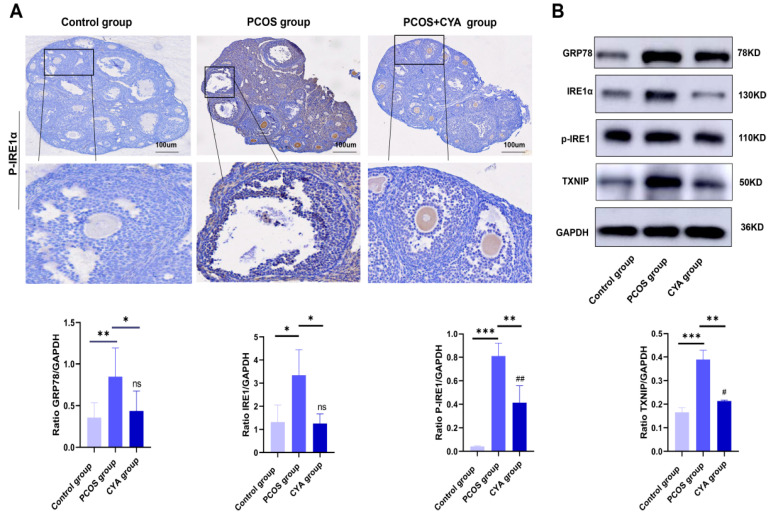
CYA effectively alleviates IRE1 signaling pathway activation induced by hyperandrogen in PCOS mice. (**A**) The expression of p-IRE1αin ovaries was assessed with immunohistochemical tests; (**B**) the expressions of IRE1α signaling pathway proteins were assessed with Western blot. Two independent experiments were performed with similar results. Data are shown as mean ± SEM. “*” represents comparison between groups, “#” indicates the treatment group compared to the control group, “ns” indicated that there was no statistical difference between the treatment group and the control group. *^/#^
*p* < 0.05, **^/##^
*p* < 0.01, *** *p* < 0.001.

**Figure 5 biology-11-01761-f005:**
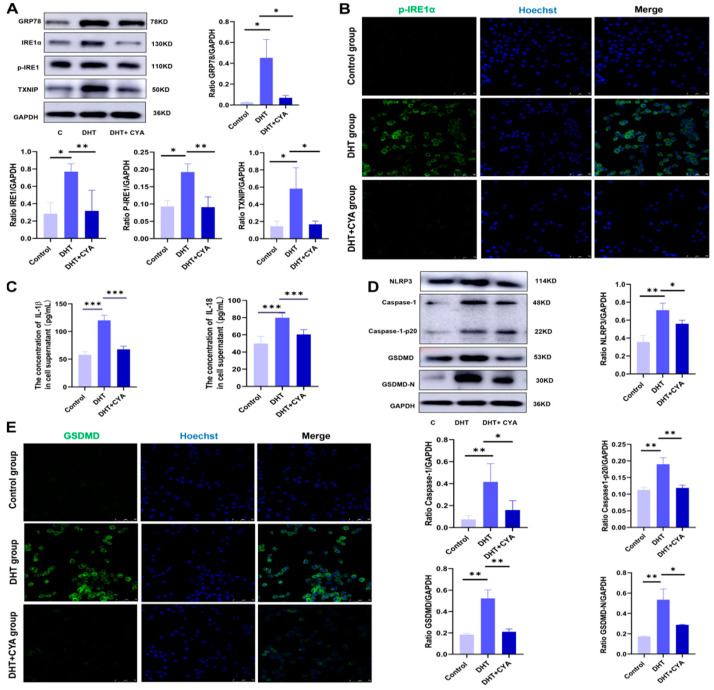
CYA could effectively alleviate the activation of the IRE1α signaling pathway and pyroptosis in ovarian GCs induced by HA in in vitro experiments. According to the preliminary experiment, the concentration of DHT was 10 uM, the action time was 24 h, and the concentration of CYA was 15 uM. (**A**) The expression of IRE1α signaling pathway protein in ovarian GCs was assessed with Western blot; (**B**) the expression of p-IRE1αin ovarian GCs was assessed with immunohistochemical tests; (**C**) the expressions of IL-18 and IL-1β in cell supernatants were analyzed using enzyme-linked immunosorbent assay kits; (**D**) the expressions of NLRP3 inflammasome and pyroptosis-related proteins in ovarian GCs were assessed with Western blot; (**E**) the expression of GSDMD in ovarian GCs was assessed with immunohistochemical tests. Two independent experiments were performed with similar results. Data are shown as mean ± SEM. * *p* < 0.05, ** *p* < 0.01, *** *p* < 0.001.

**Figure 6 biology-11-01761-f006:**
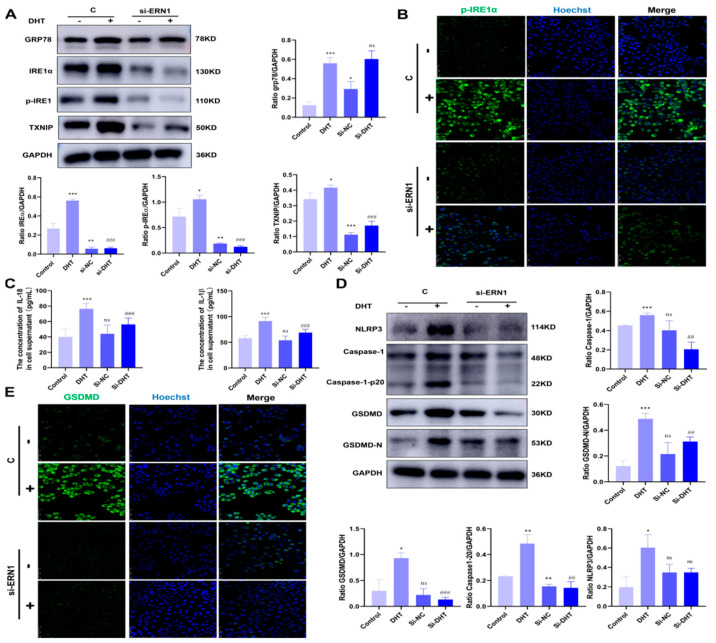
CYA can alleviate ovarian granulosa cell pyroptosis and the local ovarian inflammatory response by inhibiting the activation of the IRE1α signaling pathway in PCOS. After *ERN1* (IRE1α gene) was knocked down by siRNA in KGN cells, DHT was added to the cell culture medium, the concentration of DHT was 10 uM, and the action time was 24 h. (**A**) The expression of IRE1α signaling pathway proteins in ovarian GCs was assessed with Western blot; (**B**) the expression of p-IRE1α in ovarian GCs was assessed with immunohistochemical tests; (**C**) the expressions of IL-18 and IL-1β in cell supernatants were analyzed using enzyme-linked immunosorbent assay kits in four groups; (**D**) the expressions of NLRP3 inflammasome and pyroptosis-related proteins in ovarian GCs were assessed with Western blot; (**E**) the expression of GSDMD in ovarian GCs was assessed with immunohistochemical tests. Two independent experiments were performed with similar results. Data are shown as mean ± SEM. “*” indicates the group compared to the control group, “#” indicates the si-DHT group compared to the DHT group, * *p* < 0.05, **^/##^
*p* < 0.01, ***^/###^
*p* < 0.001.

**Figure 7 biology-11-01761-f007:**
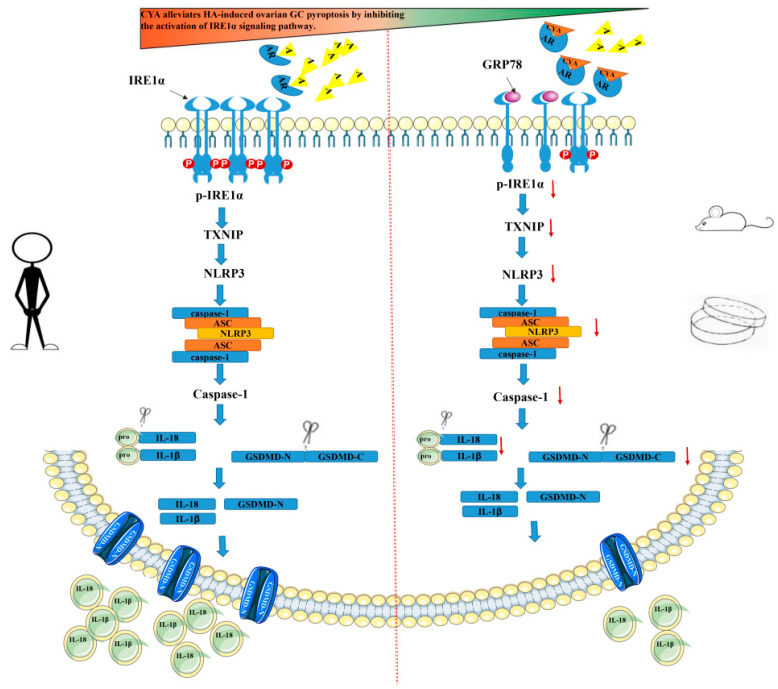
Schematic illustration of CYA alleviating ovarian granulosa cell focal collapse and local ovarian inflammatory response in PCOS by inhibiting IRE1α signaling activation. A: Androgen; AR: Androgen receptor; CYA: Cyproterone acetate; GRP78: 78-kDa glucose-regulated protein; IRE1α: Inositol-requiring enzyme type 1α; p-IRE1α: Phosphorylated Inositol-requiring enzyme type 1α; TXNIP: Thioredoxin-interacting protein; NLRP3: The nucleotide-binding oligomerization domain-like receptor family, pyrin domain-containing 3; Caspase-1: Cysteine aspartase-1; ASC: Apoptosis-associated speck-like protein containing CARD; IL-18: Interleukin 18; IL-1β: Interleukin 1β; GSDMD-C: Gasdermin-C; GSDMD-N: Gasdermin-N; “↓”indicate that expression of protein was decreased.

**Table 1 biology-11-01761-t001:** Comparison of basic clinical information and pregnancy-assisted outcomes for the two groups.

No. of Cycles	Control Group	PCOS Group	*χ*^2^/*t* Value	*p* Value
Age (years)	29.40 ± 4.34	29.73 ± 4.09	−0.306	0.761
BMI (kg/m^2^)	24.57 ± 3.89	24.33 ± 3.16	0.253	0.801
Infertility duration (years)	3.40 ± 2.19	4.37 ± 2.30	−1.668	0.101
Basal T (ng/mL)	0.29 ± 0.15	0.45 ± 0.11	−4.687	**0.000**
No. of basal antral follicles	17.73 ± 4.86	22.33 ± 4.59	−3.771	**0.000**
No. of oocytes retrieved	12.50 ± 3.85	16.33 ± 7.35	−2.529	**0.014**
No. of mature oocytes	11.03 ± 3.84	12.37 ± 5.38	−1.105	0.274
Fertilization	9.57 ± 3.54	9.53 ± 4.85	0.030	0.976
No. of available embryos	8.70 ± 3.41	7.20 ± 2.86	1.848	0.070
No. of good-quality embryos	6.80 ± 2.71	4.13 ± 1.76	4.524	**0.000**
Clinical pregnancy rate (%)	66.67(20/30)	56.67(17/30)	4.009	**0.045**
Early abortion rate (%)	10.00(2/20)	17.65(3/17)	4.930	**0.026**

Intention-to-treat analysis. Values are presented as mean (SD) unless stated otherwise. BMI: body mass index. After the data were assessed for normality by the Kolmogorov–Smirnov test, significant differences were analyzed by unpaired *t* test or chi-squared test. *p* values in bold indicate statistical significance.

## Data Availability

The data presented in this study are available on request from the corresponding author.

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
