# Peer review of "Cyproterone Acetate Mediates IRE1α Signaling Pathway to Alleviate Pyroptosis of Ovarian Granulosa Cells Induced by Hyperandrogen"

_biology, 2022, doi:10.3390/biology11121761_

Round 1

Reviewer 1 Report

In this manuscript Zhang et al investigate the effect of cyproterone acetate (CYA) on the hyperandrogenemia-induced pyroptosis of granulosa cells in PCOS. The authors examined women with PCOS (measurements in serum and follicular fluid, observation of granulosa cells under microscope), and conducted experiments on mice-models of the disease and cultured ovarian granulosa cells. They have found that the process of pyroptosis is upregulated in women with PCOS and hyperandrogenemia, and that CYA can alleviate the hyperandrogenemia-induced pyroptosis in the ovaries of mice and in cultured ovarian granulosa cells.

Overall, this is a very interesting study which I think deserves to be published. However, I think that the following points should be addressed:

1. The manuscript needs to be edited by a native English speaker, as the poor language quality makes it difficult to understand the point of the authors at various instances (for example: page 2 lines 60-63, page 7 lines 293-294).

2. Introduction, page 2 lines 83-84: CYA is commonly used for the treatment of PCOS as part of an oral contraceptive pill (as a combination with ethinyl-estradiol) but can also be used as monotherapy. I believe this should be included in your introduction.

3. Page 3 line 105: as per Rotterdam criteria, polycystic ovary appearance on ultrasound is defined as the presence of ≥ 12 follicles measuring 2‐9 mm [and not 4-9 mm] in diameter and/or an ovarian volume > 10 mL in at least one ovary.

4. Page 5 lines 237-239: I think this sentence should be omitted. What previous studies have shown should be mentioned in the introduction (and the very same information is already there!).

5. In the Results section, when describing their findings in CYA-treated mice, the authors frequently use the expression “After treatment with CYA”. Since CYA was administered in the treatment group concurrently with DHEA and not afterwards, I would suggest using the expression “in CYA-treated mice” instead. This way it will be clearer to the reader that CYA was not given to mice with already established DHEA-induced PCOS.

6. Please report if there were any significant differences regarding the serum levels of TT, IL-18, IL-1β and NLRP3, and the ovarian expression of GRP78, IRE1α, p-IRE1α and TXNIP between the control group and the CYA-treated group of mice.  

Reviewer 2 Report

Dear Authors,
you have presented interesting paper, but I have some minor comments about it.
- in section Materials and Methods there is no information about approval of the Bioethics/ Ethics Committee, add it.
- there is no information about the limitations of your study.
- the Conclusion section is too short, please indicate the practical application of your research.

Reviewer 3 Report

The reviewed manuscript in its current form does not raise any comments.
The presentation of the methods used and the results obtained is very good and can be a model for other authors.
The conclusions are fully supported by the obtained results.
The manuscript does not require corrections.
